# The Multifaceted Functionality of Plasmacytoid Dendritic Cells in Gastrointestinal Cancers: A Potential Therapeutic Target?

**DOI:** 10.3390/cancers16122216

**Published:** 2024-06-13

**Authors:** Frederik J. Hansen, Paul David, Georg F. Weber

**Affiliations:** Department of General and Visceral Surgery, Medical Faculty of Friedrich-Alexander-University Erlangen, University Hospital Erlangen, 91054 Erlangen, Germany; frederik.hansen@fau.de

**Keywords:** plasmacytoid dendritic cells, pDCs, esophageal cancer, gastric cancer, hepatocellular carcinoma, cholangiocarcinoma, pancreatic cancer, colorectal cancer

## Abstract

**Simple Summary:**

Gastrointestinal (GI) tumors present a significant global health challenge, prompting the need for new therapeutic strategies. Plasmacytoid dendritic cells (pDCs) are crucial in tumor immunity, with both anti-tumor and pro-tumor effects. This review examines pDC roles in various GI tumors and their potential as therapeutic targets. In gastric cancer, hepatocellular carcinoma, and intrahepatic cholangiocarcinoma, higher pDC infiltration correlates with worse outcomes, whereas in esophageal, pancreatic, and colorectal cancers, it improves outcomes. Overall, pDCs have a complex role in GI tumors, affecting both tumor immunity and progression. Further research is needed to refine their clinical use and explore combination therapies.

**Abstract:**

Gastrointestinal (GI) tumors pose a significant global health burden, necessitating the exploration of novel therapeutic approaches. Plasmacytoid dendritic cells (pDCs) play a crucial role in tumor immunity, exhibiting both anti-tumor and pro-tumor effects. This review aims to summarize the role of pDCs in different types of GI tumors and assess their potential as therapeutic targets. In gastric cancer, hepatocellular carcinoma, and intrahepatic cholangiocarcinoma, increased infiltration of pDCs was associated with a worse outcome, whereas in esophageal cancer, pancreatic cancer, and colorectal cancer, pDC infiltration improved the outcome. Initial animal studies of gastric cancer and hepatocellular carcinoma showed that pDCs could be a successful therapeutic target. In conclusion, pDCs play a multifaceted role in GI tumors, influencing both anti-tumor immunity and tumor progression. Further research is needed to optimize their clinical application and explore combinatorial approaches.

## 1. Introduction

Gastrointestinal (GI) tumors encompass a diverse group of malignancies arising from various organs within the digestive system, including the esophagus, stomach, liver, pancreas, and colorectum. These tumors represent a significant global health burden, contributing to high morbidity and mortality rates worldwide. They comprise 26% of total cancer cases and contribute to 35% of all cancer-related deaths [1]. The etiology of GI tumors is multifactorial, involving complex interactions between genetic predisposition, the immune system, environmental factors, and lifestyle choices [2]. While advancements in screening, diagnosis, and treatment have improved outcomes for some GI cancers, challenges persist in effectively managing these diseases, particularly in cases of advanced or metastatic disease. Moreover, the heterogeneity of GI tumors presents unique clinical and molecular characteristics, necessitating tailored approaches to diagnosis and therapy [3].

Immunotherapy has revolutionized the landscape of cancer treatment by harnessing the body’s immune system to target and eliminate cancer cells. Unlike traditional therapies such as operation, chemotherapy, and radiation, which directly target tumor cells, immunotherapy works by enhancing the immune system’s ability to recognize and destroy cancer cells. This approach capitalizes on the remarkable specificity and adaptability of the immune response, offering the potential for durable and systemic anti-tumor effects with fewer adverse effects [4]. Over the past few decades, immunotherapy has emerged as a cornerstone of cancer treatment, with remarkable success across various cancer types. From the advent of cytokine-based therapies to the more recent development of immune checkpoint inhibitors, chimeric antigen receptor (CAR)-T cell therapy, and cancer vaccines, immunotherapy has expanded the therapeutic arsenal available to oncologists and transformed the prognosis for many patients with advanced or refractory disease [5].

However, immunotherapy is not universally successful across all types of cancer. It has shown significant efficacy against what are termed “hot tumors”. These are characterized by a high inflammatory signature, extensive T cell infiltration, and the activation of checkpoints such as PD-1, CTLA4, TIM3, or Lag3 [6]. “Hot tumors” encompass head and neck cancer, kidney cancer, melanoma, and non-small-cell lung cancer. In contrast, “cold tumors”, including gastrointestinal tumors and prostate cancer, are characterized by a low inflammatory signature and a lack of intratumoral cytotoxic CD8^+^ T cells [7]. This may subsequently be associated with a poorer prognosis due to the poor response to immunotherapy. Therefore, it is crucial to deepen our understanding of tumor immunology to possibly transform these “cold tumors” into “hot tumors”.

Plasmacytoid dendritic cells (pDCs) play a pivotal role in the complex interplay between the immune system and cancer. Originally recognized for their role in antiviral immunity, pDCs have emerged as key regulators of tumor immunity and as therapeutic targets in cancer [8]. These specialized immune cells possess unique abilities to produce large quantities of type I interferons (IFNs) and to modulate immune responses within the tumor microenvironment [9]. In cancer, the role of pDCs is multifaceted. On the one hand, they can exert anti-tumor effects by promoting the activation of cytotoxic T cells and natural killer (NK) cells, thereby facilitating the elimination of cancer cells [10,11]. On the other hand, pDCs can also exhibit immunosuppressive functions, promoting tumor immune evasion and progression. For example, in breast cancer, tumor-infiltrating pDCs may contribute to immune tolerance by inducing regulatory T cells (Tregs) or producing immunosuppressive cytokines such as interleukin-10 (IL-10) [12].

The presence and function of pDCs in the tumor microenvironment have been associated with clinical outcomes in various cancer types. Increased infiltration of pDCs has been linked to both favorable and unfavorable prognoses, depending on the context and characteristics of the tumor [13]. Understanding the dynamics of pDCs in cancer and their interactions with other immune cells holds promise for developing novel immunotherapeutic strategies aimed at enhancing anti-tumor immunity and improving patient outcomes.

Therefore, the aim of this review is to summarize the role of pDCs in the different types of gastrointestinal cancers and thus to assess whether targeting pDCs represents a potential therapeutic approach.

## 2. Origin and General Function of Plasmacytoid Dendritic Cells

The origin of pDCs is located in the bone marrow. A common DC progenitor (CDP) is responsible for generating both pDCs and classical DCs (cDCs), but not other cell lineages [14,15]. This CDP expresses elevated levels of E2-2, the transcription factor that characterizes the pDC lineage. It can be derived from CDPs under conditions where E2-2 is upregulated, such as following exposure to M-CSF or thrombopoietin [16]. A notable aspect of pDC development is that, alongside the myeloid lineage as previously described, they can also arise from the lymphoid lineage. This implies that mature pDCs can develop from lymphoid-primed multipotent progenitor cells as well [17]. Despite their distinct origins, the two types of pDCs do not differ in functionality [18,19,20]. Following maturation in the bone marrow, pDCs circulate in the blood and migrate to the T cell areas of the lymph nodes via high endothelial venules, unlike cDCs, which use afferent lymphatic pathways [21,22]. In tumor diseases or inflamed tissue, pDCs can also migrate to these areas under the influence of various chemokines and exert their effects there before entering the T cell areas of the lymph nodes [23,24].

Human pDCs are phenotypically characterized through the expression of CD4, CD68, ILT3, IL-3 receptor alpha-unit (IL-3R; also known as CD123), major histocompatibility class (MHC) II, blood DC antigen 2 (BDCA-2; also known as CLEC4C), and CD304 [25,26]. They lack the expression of the lineage-associated markers CD3, CD11c, CD14, CD16, CD19 [27].

pDCs are specialized in the production of type I interferons (IFNs), particularly IFN-α and IFN-β, upon viral stimulation [28]. This ability is facilitated by their high expression of toll-like receptor (TLR) 7 and TLR9, which detect viral RNA and DNA, respectively [29]. Upon recognition of viral nucleic acids, pDCs rapidly produce large amounts of type I IFNs, which exert potent antiviral effects by inducing an antiviral state in neighboring cells, activating natural killer (NK) cells, and promoting the maturation of cDCs [30,31].

Although pDCs are less efficient in antigen presentation compared to cDCs, they still play a crucial role in initiating adaptive immune responses. pDCs can present viral antigens to CD4^+^ and CD8^+^ T cells, thereby contributing to the generation of antiviral immunity [32]. However, depending on the context, pDCs can also have a tolerogenic effect. When pDCs are activated by TLR receptors, for example, they become immunogenic cells and produce abundant type I IFNs [33]. Nevertheless, if pDCs are not stimulated or alternatively activated, they can express 2,3-dioxygenase (IDO) [34], Inducible T cell COStimulator ligand (ICOS-L) [35], Programmed Cell Death Protein 1 ligand (PD-L1) [36], or granzyme B [37], thereby inducing immunological tolerance. Consequently, the expression of IDO and ICOS-L on pDCs leads to the activation of regulatory T cells, which can effectively suppress the immune response [38,39].

A recent study showed that pDCs develop into different subpopulations with distinct phenotypic and molecular characteristics after a single stimulus. They are specialized in the production of type I IFN (P1-pDCs), in adaptive immune function (P3-pDCs), or in both (P2-pDCs). P1-pDCs are PD-L1^+^CD80^−^, P3-pDCs are PD-L1^−^CD80^+^, and P2 express both surface markers [40]. However, studies investigating the subtypes of pDCs in various tumor diseases are still lacking.

pDCs also play a significant role in mucosal immunity. Interestingly, unlike splenic pDCs, pDCs from the Peyer’s patches cannot produce type I IFNs after TLR7 or TLR9 stimulation [41]. However, these pDCs still produce pro-inflammatory cytokines that can trigger a T cell response [42]. Moreover, mucosal pDCs can initiate a T cell-independent B cell response [43]. Thus, the expression of B cell-activating factor (BAFF) and proliferation-inducing ligand (APRIL) on pDCs leads to IgA production in the mucosa [44]. This highlights that pDCs can exhibit different functionalities in various body compartments, which is crucial for understanding their roles in different gastrointestinal tumors.

## 3. Anti-Tumoral Effects of pDCs

pDCs possess the capability to trigger both innate and adaptive immune responses against tumor cells. For instance, they can demonstrate direct cytotoxic effects on tumor cells by releasing granzyme B and tumor necrosis-related apoptosis-inducing ligand (TRAIL). Granzyme B is a serine protease that induces apoptosis and emerging extracellular functions, including the cleavage of cell–cell junctions, cell receptors, and basements membranes [45,46]. TRAIL is a molecule belonging to the tumor necrosis factor superfamily (TNF) [47]. Tumors express the TRAIL receptor and can signal apoptosis upon binding of TRAIL [48]. While pDCs typically exhibit minimal secretion of these cytotoxic agents in their resting state, the activation of toll-like receptors (TLRs) was demonstrated to enable them, in ex vivo studies, to effectively eliminate tumor cells originating from hematological malignancies, breast cancer, and melanoma [49,50,51].

pDCs are recognized as the primary source of IFN-α, a cytokine possessing significant anti-tumor properties. Of particular significance is its role in activating T cells and NK cells, both proficient in eliminating cancerous cells [52]. Moreover, IFN-α stimulates the presentation of tumor antigens, thereby amplifying the adaptive immune response [53]. Additionally, IFN-α exerts a direct impact on tumor cells by impeding their growth and migration [54,55].

pDCs serve as antigen-presenting cells. Following activation, they upregulate MHC and T cell co-stimulatory molecules. Upon uptake of the tumor antigen, they can migrate to lymphoid organs and present antigens via MHC I and II to CD8^+^ and CD4^+^ T cells [56]. This process initiates an adaptive immune response against the tumor.

In addition to IFN-α, pDCs are capable of producing TNF-α. This leads to increased processing and presentation of tumor antigens, consequently amplifying T cell activation [57]. Interestingly, TNF-α-producing pDCs exhibit altered functionality: they scarcely produce IFN-α and display increased expression of MHC I and II, along with the co-stimulatory molecules CD80, CD86, and CCR7 [58]. This underscores the complexity of pDC functionality, which may manifest differently across various cancer types or tumor stages. Figure 1 schematically depicts the anti-tumoral functions of pDCs.

## 4. Pro-Tumoral Effects of pDCs

pDCs are capable of entering a tolerogenic state, thereby contributing to an immunosuppressive environment. For instance, a tumor may produce transforming growth factor-ß (TGF-ß) and prostaglandin E2 (PGE2). Consequently, the production of IFN-α and TNF-α in pDCs is significantly diminished, rendering the anti-tumor functions of these cytokines ineffective [59].

pDCs can secrete some tumor-promoting molecules. For instance, one study demonstrated that CD40-dependent activation of pDCs resulted in the induction of neoangiogenesis through the secretion of IL-8 and TNF-α in the ascites of ovarian cancer patients [60]. Additionally, tumor-infiltrating pDCs in non-small-cell lung cancer produced IL-1α, a proangiogenic and proinvasive cytokine [61]. As mentioned above, pDCs can secrete granzyme B, which can eliminate tumor cells through its cytotoxic effect. However, on the other hand, granzyme B is also capable of inhibiting the proliferation of CD4^+^ and CD8^+^ T cells, thereby favoring tumor growth [37].

Moreover, tolerogenic pDCs express ICOS-L, inducing naive CD4^+^ T cells to differentiate into IL-10-producing Tregs [62,63]. Additionally, tumor-infiltrating pDCs can secrete IDO, resulting in the activation of Foxp3^+^ Tregs [34]. This T cell subtype provides the main mechanism of tumor immune evasion: they directly interact with other immune cells, such as T cells, dendritic cells, and macrophages, through cell-to-cell contact [64,65,66]. Tregs express inhibitory molecules, such as Cytotoxic T-Lymphocyte-Associated Protein 4 (CTLA-4) and Programmed Cell Death Protein 1 (PD-1), which bind to their respective ligands on the target cells, delivering inhibitory signals that dampen their activation and function through inducing apoptosis [67,68]. Furthermore, Tregs produce immunosuppressive molecules, such as IL-10 and TGF-ß. These molecules inhibit the activation, proliferation, and effector functions of CD4^+^ and CD8^+^ T cells, thus suppressing the anti-tumoral T cell response [69,70].

In addition to ICOS-L and all the described immunosuppressive molecules, tolerogenic pDCs express PDL-1 themselves, further facilitating tumor immune escape [37,71].

In breast cancer, for example, it was demonstrated that increased infiltration of pDCs is associated with a poorer outcome [72,73]. Additionally, tumor-infiltrating pDCs exhibited minimal production of IFN-α but instead showed increased expression of ICOS-L on their surface. This resulted in the activation of IL-10-producing Tregs and, consequently, tumor progression [12,74]. Similarly, in ovarian tumors, pDCs exhibited minimal IFN-α production but induced IL-10 production by naïve CD4^+^ T cells [35]. Figure 2 schematically summarizes several pro-tumoral functions of tolerogenic pDCs.

## 5. Plasmacytoid Dendritic Cells in Various Gastrointestinal Cancers

### 5.1. Esophageal Cancer

Esophageal cancer is the sixth leading cause of death from cancer and the eighth most common cancer in the world [75]. However, little is known about the role of plasmacytoid dendritic cells in this cancer type. A comprehensive investigation employing RNA analysis of tumor tissue in esophageal squamous cell carcinoma (ESCC) unveiled the infiltration of pDCs in both early-stage and advanced tumors. Patients characterized by heightened pDC infiltration exhibited significantly enhanced overall survival (OS) rates in contrast to those exhibiting diminished infiltration levels [76]. Our preliminary data show a significant decrease in the frequencies of pDCs in the peripheral blood of preoperative esophageal adenocarcinoma and ESCC patients compared to clinical control patients. However, due to the lack of further studies, it is currently impossible to determine the functionality of pDCs and their potential therapeutic application in esophageal cancer. Further studies are necessary to address this gap in knowledge. Table 1 summarizes the clinical impact of increased pDC infiltration in various gastrointestinal tumors.

### 5.2. Gastric Cancer

Even more common than esophageal cancer is gastric cancer (GC), which ranks as the fifth most common cancer and the third most common cause of cancer death worldwide [86]. There are several risk factors for the development of gastric cancer, such as smoking, high salt intake, and alcohol consumption [87].

In this context, chronic inflammation of the gastric mucosa induced by *Helicobacter pylori* (*H. pylori*) plays a pivotal role in initiating the progression towards gastric cancer [88]. Interestingly, pDCs from H. pylori-infected patients already showed alterations. Patients infected with *H. pylori* exhibited elevated HLADR expression on circulating pDCs compared to those without *H. pylori* infection [89]. Additionally, a higher density of pDCs was observed in the gastric mucosa [89]. Moreover, it was demonstrated that within the tumor microenvironment of GC, there is a positive correlation between *H. pylori* infection and the expression of TLR9 and ICOS-L on pDCs [90]. This correlation, in turn, is associated with the expression of ICOS on tumor infiltrating Tregs [90]. This suggests that pDCs are involved in chronic inflammation by *H. pylori* and thus possibly contribute to the genesis of GC [91].

Regardless of the cause of carcinogenesis, an increased number of pDCs was found in the peripheral blood of GC patients compared to healthy individuals [92,93]. The increased number of circulating pDCs in GC patients correlated with advanced stages and the presence of lymphatic metastases [93]. Interestingly, despite the increased number of circulating pDCs, IFN-α concentration in plasma was lower in GC patients compared to healthy individuals [92]. This would indicate that pDCs change their functionality in the presence of cancer. Furthermore, it was found that the number of circulating pDCs correlated positively with the number of ICOS^+^ Tregs in the blood of GC patients [92]. Both ICOS^+^ Tregs and pDCs in the blood of GC patients predicted poorer overall survival [77].

Also, tumor tissue exhibited heightened infiltration of pDCs in comparison to normal tissue [77,79,94]. A higher number of pDCs within the tumor tissue correlated with reduced overall survival [77,78], larger tumor size [79], and greater tumor depth of invasion [80]. Plasmacytoid dendritic cells also seem to play a role in promoting tumor progression within the tumor tissue by inducing Tregs. Thus, pDC infiltration correlated positively with ICOS^+^ Treg infiltration in the tumor tissue of GC patients [92]. Both predicted poor clinical outcome in GC patients [77]. In advanced tumor stages, there was notably increased infiltration of ICOS^+^ Tregs [90,92]. Interestingly, ICOS^+^ Tregs were found mainly distributed in the carcinoma tissue, whereas pDCs were mainly located in peritumoral tissue. A similar distribution was also observed for cervical cancer [95] and intrahepatic cholangiocarcinoma [83].

There have already been initial attempts to reactivate tolerogenic pDCs through vaccination, thereby inducing increased IFN-α production, which can then unleash their anti-tumor properties. In a mouse model of GC, vaccination with a TLR7 agonist conjugated with a GC tumor antigen resulted in a reduction in tumor size by inducing a cytotoxic T cell response [96]. In another mouse model, this vaccination was combined with the most common chemotherapeutic agent, 5-fluorouracil (5-FU). This combination proved to be particularly advantageous, as the addition of 5-FU inhibited immunosuppressive myeloid-derived suppressor cells (MDSCs), thereby strengthening the T cell response [97].

In summary, it can be said that pDCs undeniably play a significant role in the progression of GC. pDC-based vaccination appears to offer a potential therapeutic benefit. However, further studies are necessary to test this in humans.

### 5.3. Liver Malignancies

#### 5.3.1. Hepatocellular Carcinoma (HCC)

Hepatocellular carcinoma (HCC) is the most prevalent liver cancer and, like many other gastrointestinal tumors, still carries a poor prognosis due to insufficient therapy [98]. In the blood, HCC patients showed a reduced number of pDCs, while these were increased in ascitic fluid [82] and the tumor tissue [99,100]. Furthermore, the number of intratumoral pDCs correlated with the number of intratumoral Tregs [81,100,101]. pDCs exposed to tumor tissue stimulate Tregs to increase IL-10 production by up-regulating the expression of ICOS-L [100]. This exerts an immunosuppressive effect, thereby contributing to cancer progression through immune evasion. This is consistent with the finding of another study demonstrating that the infiltration of pDCs in tumors correlated with high α-fetoprotein levels and greater vascular und lymphatic invasion after curative resection. This resulted in poorer overall survival and shorter time to recurrence [81,82]. A statistical analysis indicated that pDC infiltration was an independent predictor of overall survival and time to recurrence in HCC patients [81]. Interestingly, this correlation was found in the tumor tissue and not in the peritumoral tissue, as described in gastric cancer. 

Sorafenib is a multikinase inhibitor that slows the growth of HCC via apoptosis, cell cycle arrest, and anti-angiogenesis mechanisms [102]. It is used as standard therapy in advanced stages of HCC [103]. However, it also affects immune cells, exerting a regulatory impact on T cells [104], NK cells [105], macrophages [106], and dendritic cells [107]. This severely impairs anti-tumor immunity. Regarding the impact on pDCs, sorafenib-treated HCC patients produced less IFN-α then untreated patients [108]. Furthermore, ex vivo analyses showed that in pDCs of HCC patients, the application of sorafenib significantly reduced IFN-α production [108]. However, sorafenib primarily exerts its potent effect directly on tumor cells, resulting in a relatively modest influence on the immunogenic or tolerogenic status of pDCs.

HCC cells also contribute to the immunosuppressive environment. For example, they express the ectonucleotidases CD39 and CD73, which generate the immunosuppressive metabolite adenosine [109,110]. Adenosine leads to increased migration of pDCs into the tumor through the adenosine A1 receptor (ADORA1), thus inducing Tregs and consecutively suppressing cytotoxic CD8^+^ T cells [82]. In a mouse model of HCC, it was demonstrated that the depletion of pDCs led to a suppression of tumor growth by reducing the infiltration of Tregs and increasing the infiltration of cytotoxic CD8^+^ T cells [82]. Since complete depletion of pDCs could lead to immune-related adverse effects, only ADORA1 of pDCs was blocked in a subsequent mouse model. In that model, tumor growth could also be inhibited [82]. This indicates that pDCs are a promising potential therapeutic target in HCC. Clinical trials are now essential to investigate the potential benefits in humans as well.

#### 5.3.2. Intrahepatic Cholangiocarcinoma (ICC)

Intrahepatic cholangiocarcinoma (ICC) is the second most common liver cancer after HCC with a globally increasing incidence [111,112]. Its high recurrence rate shortens the long-term survival of patients [113,114]. In ICC patients who underwent primary curative resection, the number of peritumoral pDCs correlated positively with tumor size, vascular and bile duct invasion, lymphatic metastasis, and TNM (tumor, node, and metastasis) stage [83]. The statistical analysis revealed that the infiltration of pDCs is an independent predictor for the time to recurrence (TTR) and, thus, also for overall survival in ICC patients [83]. As a potential underlying mechanism, the study demonstrated a positive correlation between peritumoral pDC infiltration and the infiltration of Foxp3^+^ regulatory T cells [83]. To investigate not only the described prognostic benefit but also a potential therapeutic approach involving pDCs for ICC, additional studies are required.

### 5.4. Pancreatic Cancer (PDAC)

Pancreatic ductal adenocarcinoma (PDAC) exhibits one of the poorest prognoses among all gastrointestinal tumors [115]. This is attributed to its late diagnosis, ability to rapidly form metastasis, challenging anatomical location, aggressive tumor growth, and nearly complete resistance to immunotherapy [116]. Hence, it is crucial to explore novel therapeutic strategies for this condition. It is surprising, however, that pDCs are rarely investigated in the context of this cancer. It was demonstrated that PDAC patients exhibit lower levels of circulating pDCs compared to control patients. Furthermore, these pDCs displayed an increased rate of apoptosis [117]. In PDAC, pDCs also infiltrate the peritumoral area and are significantly less abundant intratumorally [84]. This only tended to show that PDAC patients with an increased infiltration of pDCs had an improved disease-free survival (DFS). Nevertheless, a statistical analysis revealed that pDC infiltration is an independent prognostic factor for DFS but not for overall survival [84]. Our unpublished data from preoperative PDAC patients demonstrate a significant decrease in the frequency of circulating pDCs in these patients compared to clinical control patients. Larger-scale studies are urgently required to gain a deeper understanding of the precise functionality of pDCs in this devastating cancer.

### 5.5. Colorectal Cancer (CRC)

Colorectal carcinoma (CRC) is the most common cancer among gastrointestinal tumors. Intensive research has significantly improved its prognosis in recent decades [118]. In this cancer, pDC frequencies were significant lower in tumor tissue compared to normal mucosa [119]. In rectal carcinoma, increased infiltration of pDCs was observed compared to colon carcinoma [120]. This may be attributed to the fact that neoadjuvant therapy increases the number of pDCs [120], and rectal carcinomas are routinely subjected to this therapy when certain risk criteria are present.

Increased infiltration of pDCs in CRC has been associated with smaller tumor size, reduced lymphatic metastases, decreased distant metastases, and consequently, lower TNM stage. Therefore, CRC patients with increased infiltration of pDCs showed improved overall survival and progression-free survival [85]. Interestingly, tumor-infiltrating pDCs showed increased expression of interferon regulatory factor 7 (IFR7) [85]. The upregulation of this gene leads to increased IFN-α production [121], thus enabling pDCs to inhibit the growth of CRC cells through the anti-tumoral properties of IFN-α. Additional evidence supporting the tumor-combating capabilities of pDCs is their positioning within the tumor stroma of CRC, in close proximity to cytotoxic CD8^+^ T cells that produce granzyme B [85]. Additionally, a high expression of CLEC4C genes in pDCs was associated with better survival in CRC [120]. CLEC4C is responsible for the production of the BDCA-2 protein, which plays a crucial role in ligand internalization and antigen presentation [122,123]. An additional indication that pDCs also serve an antigen-presenting function in CRC is their localization within the tertiary lymphatic structure of the tumor, directly adjacent to CD4^+^ T cells [85].

Interestingly, several studies demonstrated that an increased infiltration of Tregs is associated with a better outcome in CRC [124,125]. It is plausible that Tregs decelerate chronic inflammation, thereby diminishing the oncogenic potential. Presently, no studies have established a correlation between pDCs and Tregs in CRC. However, this underscores once again the considerable differences in tumor biology among various gastrointestinal tumors.

In summary, pDCs in CRC possess the capability to initiate an anti-tumor T cell response against the tumor.

## 6. pDC-Based Cancer Treatment

In “hot tumors” like melanoma, initial studies have already been conducted in humans using pDCs as a therapeutic target. There is a particular focus on DC-based vaccination to bolster the anti-tumor T cell response. It was already demonstrated in a small cohort of metastatic melanoma patients that the administration of natural pDCs is safe and triggers an antigen-specific CD4^+^ and CD8^+^ T cell response [126,127]. Here, pDCs exhibited a significant difference in cytokine release compared to cDCs. This resulted in an increased influx of T cells with cytolytic activity into the skin [128]. cDCs and pDCs were demonstrated to mutually activate each other and boost the NK cell-mediated destruction of tumor cells. This indicates that merging different human blood DC subsets could potentially heighten the effectiveness of anticancer vaccines [129]. Another approach involves activating pDCs using a virus. Ex vivo experiments demonstrated that vaccination with an attenuated measles or herpes simplex 1 virus results in the maturation of pDCs. Subsequently, these cells produce a substantial amount of IFN-α and present melanoma tumor antigens to CD8^+^ T cells. Moreover, they activate NK cells and are capable of directly killing melanoma tumor cells through the production of granzyme B and TRAIL [130,131]. Table 2 summarizes different approaches of pDC-based cancer treatments in humans.

As previously described, IFN-α possesses certain anti-tumor properties. The U.S. Food and Drug Administration (FDA) approved the application of IFN-α for hairy cell leukemia and advanced melanoma [135]. However, alongside its moderate effectiveness, severe side effects including fever, diarrhea, psychosis, liver toxicity, thrombocytopenia, and leukopenia are also observed [136,137,138,139]. Therefore, the approach of inducing tolerogenic pDCs to reproduce IFN-α through the application of TLR agonists was pursued. The application of a TLR9 agonist in the skin of melanoma patients resulted in local regression by initiating a CD8^+^ T cell response [132,140]. In a mouse model of head and neck squamous cell carcinoma (HNSCC), the combination of a TLR9 agonist and a PD-1 blocker demonstrated tumor growth regression, leading to prolonged survival compared to PD-1 administration alone. This anti-tumor effect resulted from the increased secretion of IFN-γ from CD4^+^ and CD8^+^ T cells that were previously activated by pDCs [133]. It would be intriguing to explore this approach in gastrointestinal tumors as well, given that immune blockers are approved for metastasis treatment in most GI tumors.

There are also initial approaches targeting pDCs in humans for “cold tumors”. For instance, the application of mature pDCs in prostate cancer patients showed an increased number of IFN-γ-producing antigen-specific T cells in skin biopsies, and this also correlated with progression-free disease [134]. Further studies are urgently needed to pursue this approach in gastrointestinal tumors as well. In addition, efforts should continue to find ways to convert “cold tumors” into “hot tumors”, so that immunotherapy can be more effective.

## 7. Conclusions

In summary, it can be stated that pDCs exhibit diverse functions across various gastrointestinal tumors. Nonetheless, pDCs have proven to be effective therapeutic targets in mouse models of GC and HCC. Therefore, they may also hold promise as therapeutic targets for other types of tumors. However, since gastrointestinal tumors are cold tumors, it is questionable whether targeting pDCs alone will be successful in tumor therapy. Rather, investigations into combinational therapeutic strategies involving pDC targeting, alongside conventional chemotherapy or immunotherapy, in gastrointestinal cancers are warranted.

## Figures and Tables

**Figure 1 cancers-16-02216-f001:**
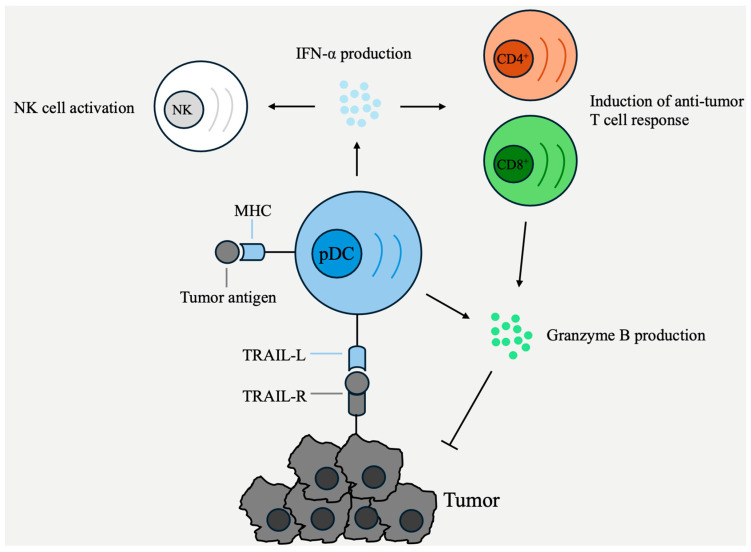
The functionality of pDCs in aiding the eradication of cancerous cells.

**Figure 2 cancers-16-02216-f002:**
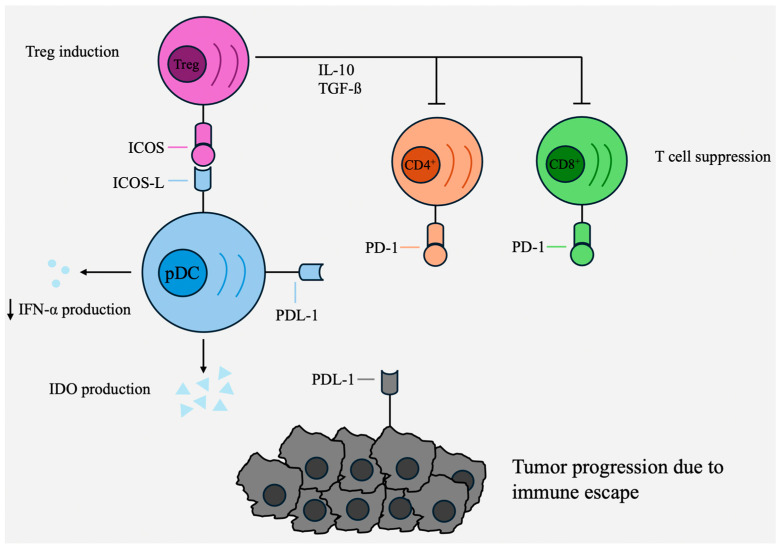
The role of tolerogenic pDCs in contributing to the evasion of tumors from immune system surveillance.

**Table 1 cancers-16-02216-t001:** Summary of the clinical impact of increased pDC infiltration in gastrointestinal tumors. OS: overall survival; TTR: time to recurrence; DFS: disease-free survival.

Cancer Type	Clinical Impact of Increased pDC Infiltration	Reference
Esophageal cancer	Improved OS	[76]
Gastric cancer	Worse OS	[77,78]
	Larger tumor size	[79]
	Deeper tumor depth of invasion	[80]
Hepatocellular carcinoma	Worse OS	[81,82]
	Shortened TTR	
	Vascular and lymphatic invasion	
	Higher α-fetoprotein levels	
Intrahepatic cholangiocarcinoma	Worse OS	[83]
	Shortened TTR	
	Larger tumor size	
	Vascular and bile duct invasion	
	Higher TNM stage	
Pancreatic cancer	Tendentially improved DFS	[84]
Colorectal Cancer	Improved OS and DFS	[85]
	Smaller tumor size	
	Reduced lymphatic invasion	
	Less metastasis	
	Lower TNM stage	

**Table 2 cancers-16-02216-t002:** Overview of pDC-based cancer treatment.

Cancer Type	Therapeutic Target	Observed Immune Response	Reference
Melanoma	Administration of natural pDCs	Induction of CD4^+^ and CD8^+^ T cell response	[126,127,128]
	Viral activation of pDCs	IFN-α production and antigen presentation	[130,131]
	Administration of TLR9 agonist	Induction of CD8^+^ T cell response	[18,132]
Head and neck cancer (HNSCC)	Administration of TLR9 agonist	pDC-mediated T cell activation	[133]
Prostate cancer	Administration of natural pDCs	Increased IFN-γ-producing antigen-specific T cells	[134]

## Data Availability

For all data requests, please contact G.F.W.

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
