# Peer review of "The Multifaceted Functionality of Plasmacytoid Dendritic Cells in Gastrointestinal Cancers: A Potential Therapeutic Target?"

_cancers, 2024, doi:10.3390/cancers16122216_

Round 1
Reviewer 1 Report
Comments and Suggestions for Authors
This review summarized the role of plasmacytoid dendritic cells in different types of gastrointestinal tumors and assesses their potential as therapeutic targets, including gastric cancer, hepatocellular carcinoma, and intrahepatic cholangiocarcinoma. Increased infiltration of plasmacytoid dendritic cells was associated with a worse outcome, whereas in esophageal cancer, pancreatic cancer, and colorectal cancer, plasmacytoid dendritic cells infiltration improved the outcome. Plasmacytoid dendritic cells play a multifaceted role in GI tumors. Therefore, the authors concluded that further research is needed to optimize their clinical application and explore combinatorial approaches. This is a well-written manuscript. I recommend accepting this paper for publication after minor revisions.
1. Line 225, 227, 228, 229, 231, 234. “Helicobacter pylori” or “H. pylori” should be italics.
2. Line 239. 377. “Ex-vivo” should be italics.
3. Please give the full name of “TNM stage”.
Author Response
We thank the reviewer for this positive feedback. We have accepted the suggested improvements and italicized "H. pylori" and "ex-vivo." Additionally, we have spelled out TNM at its first mention to ensure clarity for the reader.
Reviewer 2 Report
Comments and Suggestions for Authors
The review article presented by Frederik J. Hansen et al. focuses on the role of plasmacytoid dendritic cells (pDCs) in the different types of gastrointestinal (GI) tumors and assesses their potential as therapeutic targets. pDCs play a crucial role in tumor immunity, exhibiting both anti-tumor and pro-tumor effects. In gastric cancer, hepatocellular carcinoma, and intrahepatic cholangiocarcinoma, increased infiltration of pDCs was associated with a worse outcome, whereas in esophageal cancer, pancreatic cancer, and colorectal cancer, pDC infiltration improved the outcome. First the authors illustrate the origin and general function of pDCs. A distinctive trait between pDCs and conventional DCs is their ability to produce high amount of IFN- α when stimulated. They discuss that pDCs can exhibit different functionalities in various body compartments, which is crucial for understanding their roles in different gastrointestinal tumors. Then they summarize anti-tumoral and pro-tumoral effects of pDCs., and describe the presence of pDCs in various gastrointestinal cancers.
The authors conclude that pDCs play a multifaceted role in GI tumors, influencing both anti-tumor immunity and tumor progression. Further research is needed to optimize their clinical application and explore combinatorial approaches.
The review is interesting and includes a balanced, comprehensive and critical view of the research area. It is well written and easy to read. Figures and table summarizing the clinical impact of increased pDC infiltration in various gastrointestinal tumors help the reader to better follow the text.
Minor points:
1. An additional table showing pDCs based cancer treatment should be helpful for the reader.
2. 5.3.2. Intrahepatic cholangiocarcinoma (ICC), line 306; 5.4. Pancreatic cancer (PDAC) line 318, 5.5. Colorectal cancer (CRC) line 335, should be written in bold
3. Ref. 13, 36, 45, 89, 108, 119, 120, 139: number of pages is missing
Author Response
We sincerely thank you for this positive and detailed feedback. We have gladly accepted the suggestions and implemented a table for the pDC-based cancer therapy. Additionally, we have added the missing page numbers for certain references. We have also bolded the headings as suggested.